# Comparative Proteomic Analysis Reveals Immune Competence in Hemolymph of *Bombyx mori* Pupa Parasitized by Silkworm Maggot *Exorista sorbillans*

**DOI:** 10.3390/insects10110413

**Published:** 2019-11-18

**Authors:** Ping-Zhen Xu, Mei-Rong Zhang, Li Gao, Yang-Chun Wu, He-Ying Qian, Gang Li, An-Ying Xu

**Affiliations:** 1School of Biotechnology, Jiangsu University of Science and Technology, Sibaidu Rd, Zhenjiang 212018, China; zmr198254@163.com (M.-R.Z.); gaol910@163.com (L.G.); jkdwyc@163.com (Y.-C.W.); qianheying123@163.com (H.-Y.Q.); sdlig-1@163.com (G.L.); 2Sericulture Research Institute, Chinese Academy of Agricultural Sciences, Sibaidu Rd, Zhenjiang 212018, China

**Keywords:** *Bombyx mori*, *Exorista sorbillans*, parasitoid, proteomics, two-dimensional gel electrophoresis

## Abstract

The silkworm maggot, *Exorista*
*sorbillans*, is a well-known larval endoparasitoid of the silkworm *Bombyx*
*mori* that causes considerable damage to the silkworm cocoon crop. To gain insights into the response mechanism of the silkworm at the protein level, we applied a comparative proteomic approach to investigate proteomic differences in the hemolymph of the female silkworm pupae parasitized by *E*. *sorbillans*. In total, 50 differentially expressed proteins (DEPs) were successfully identified, of which 36 proteins were upregulated and 14 proteins were downregulated in response to parasitoid infection. These proteins are mainly involved in disease, energy metabolism, signaling pathways, and amino acid metabolism. Eight innate immune proteins were distinctly upregulated to resist maggot parasitism. Apoptosis-related proteins of cathepsin B and 14-3-3 zeta were significantly downregulated in *E*. *sorbillans*-parasitized silkworm pupae; their downregulation induces apoptosis. Quantitative PCR was used to further verify gene transcription of five DEPs, and the results are consistent at the transcriptional and proteomic levels. This was the first report on identification of possible proteins from the *E*. *bombycis*-parasitized silkworms at the late stage of parasitism, which contributes to furthering our understanding of the response mechanism of silkworms to parasitism and dipteran parasitoid biology.

## 1. Introduction

To combat microbial infection and eukaryotic parasite infestation, insects have efficient and potent innate immune systems [1,2,3,4,5,6]. Infection and infestation activate defense mechanisms, including cellular and humoral immune responses. Exogenous invaders are first recognized by recognition factors to trigger cellular immune reactions and humoral immune reactions [2,4]. After the recognition procedure, modulating and signaling factors are next activated [2]. Signaling transduction is stimulated in major immune tissues, such as the fat body and hemocytes, and the gene encoding effectors are activated through signaling cascades [2,4]. The effector molecules are produced in specific tissues and secreted into the hemolymph [2].

The domesticated silkworm originated in China, has been distributed to different parts of the world [7], and is an economically important animal. The silkworm maggot *Exorista sorbillans*, a well-known larval endoparasitoid of the silkworm, is found in all silk producing areas of Asia, severely damaging the silkworm cocoon crop [8,9]. The mated gravid female *E*. *sorbillans* oviposit on the epidermis of a silkworm larva. After about 48 hours, the parasitic maggots hatch and invade the *Bombyx mori* larvae. In silkworms, immune reactions are triggered after infestation by *Exorista bombycis* [2,8,9]. The expression of immune proteins and the encoding genes are enhanced in the epithelium of *E*. *bombycis*-infested silkworm larvae. In hemocytes of the host *B*. *mori* larva after infestation by the parasitoid larva of *E*. *bombycis*, the level of reactive oxygen species (ROS) as measured by H_2_O_2_ production increases from six hours and continues to increase, significantly reaching maximum at 48 h; the H_2_O_2_ production causes cytotoxicity, lipid peroxidation, and membrane porosity that suppress both the humoral- and cell-mediated immune responses of hemocytes in *B*. *mori* [10]. In the early stage of parasitization, the antioxidative enzyme levels are maintained at a high level in silkworm hemocytes, revealing the continuous need for antioxidative enzymes to prevent immune suppression by enduring parasitism in the host [11]. Under parasitic influence, the expressions of cell apoptosis-associated genes, including autophagy 5-like (*Atg5*), apoptosis-inducing factor (*AIF*), and nedd2-like caspase, are enhanced in the larval integument of *B*. *mori* [12], which indicates parasitism-induced activation of apoptosis in the host [8,9].

At a stable temperature of 25 °C, the eggs of silkworm maggot, *E*. *sorbillans* are incubated for about two days before hatching, after which the newly hatched first instar parasitoid larvae invade the host *B*. *mori* cuticle. Maggots invade the silkworm body and parasitize between the body wall and muscle, which is identified by the presence of black markings on the epidermis at the point of infection [9]. The parasitoid maggot completes larval stages inside the silkworm fifth instar larvae for about five days [9]. Some molecular mechanisms of the early stage of parasitization have been reported previously, but the mechanism during the late stage is not yet clear. Otherwise, during silkworm larval–pupal metamorphosis, degradation of tissues that are no longer needed is an essential process [13]. *B*. *mori* undergo distinct morphological changes; the 9th and 12th ventral segments of larva are healed in pupa, which may impact maggot parasitism.

In this study, we used two-dimensional gel electrophoresis (2-DE) combined with mass spectrometry (MS) to explore the differences in hemolymph protein expression in the day-1 female silkworm pupae parasitized by *E*. *sorbillans*. This is the late stage and the third (final) instar larva of the endoparasitic maggot. Then, the third instar maggot was elicited from the silkworm body and pupated outside. We successfully identified 50 differentially expressed proteins (DEPs), obtained protein information, and annotated the molecular function. Our study provides an overview of the proteomic profile in the hemolymph of *B*. *mori* response to *E*. *sorbillans* parasitic infection and lays a foundation for clarifying the mechanism of silkworm resistance to *E*. *sorbillans*.

## 2. Materials and Methods

### 2.1. Experimental Animals and Sample Preparation

Larvae of the *B. mori* strain Baiyu were reared on mulberry leaves at a stable temperature of 25 °C. The day-2 fifth instar larvae were exposed to mated gravid females of *E. sorbillans* for oviposition for 3 hours. Only one egg was allowed on the larval surface of each host through physical removal of other eggs if any were present. Control larvae were maintained without infestation. The infected and control silkworm larvae were fed until the silkworm matured to avoid the effect of starvation. The larvae, larvae–pupae, and pupae were maintained under a 12-h light/dark photoperiod at 25 °C and 70% humidity. Hemolymph, hemocytes, and the fat body were collected from the infected and control female silkworm pupae on the first day of pupation. This time is the late stage and the third (final) instar larva of the endoparasitic maggot. Then, the third instar maggot was obtained from the silkworm body and pupated outside. The hemolymph samples were centrifuged for 10 minutes at 12,000 rpm at 4 °C and stored in a lysis buffer of 9 M urea, 4% the zwitterionic 3-[(3-cholamidopropyl) dimethylamino]-1-propanesulfonate (CHAPS), 1% dithiothreitol, 1% immobilized pH gradient (IPG) buffer, and a 1% protease inhibitor cocktail. The total protein content was quantified using a Bradford assay kit (Bio-Rad, Hercules, CA, USA). The hemolymph is an open circulatory system. The major proteins in the hemolymph are produced in other specific tissues and secreted into the hemolymph, and the fat body is the one of the specific tissues [14]. Thus, the hemocytes and fat body samples were used to verify gene expressional analysis.

### 2.2. 2-DE and Protein Digestion

The proteins were separated with 2-DE. In short, 200 µg of each sample protein was added to a 24-cm broad range IPG strip (nonlinear, pH 3 to 10) for isoelectric focusing (IEF), and 2-DE was performed in 12.5% polyacrylamide gel. The gel was stained with silver nitrate following the 2-DE procedure [15]. Spots were scanned at 300 dpi using a high-resolution image scanner and analyzed using PDQuest 8.0 software (Bio-Rad, Hercules, CA, USA). For statistical analysis of the data, we used a Student’s *t*-test and the fold ratio was calculated. Three replicates were performed, and a threshold of *p* ≤ 0.05 and fold changes of ≥2.5 or ≤0.4 were used to identify differently expressed protein spots. The marked protein spots were identified. Differentially expressed protein spots were cut from the gel with a scalpel and washed twice with ultrapure water. The samples were destained for 5 min, the destaining solution was removed, and the samples were washed twice and incubated in 50% acetonitrile for 5 min, removing the acetonitrile with the addition of 100% acetonitrile for 5 min. Each sample was rehydrated in 4.0 μL of trypsin solution (Promega, Madison, WI, USA) for 30 min, and we added 16 μL of cover solution. After digestion at 37 °C for 16 h, the supernatant was transferred into a new tube and extracted once with 50 μL extraction buffer (67% acetonitrile and 5% trifluoroacetic acid). The combined extraction solution was completely dried. The dried peptides were dissolved in 5 μL 0.1% trifluoroacetic acid (TFA) and then mixed in a 1:1 ratio with a saturated solution of α-cyano-4-hydroxycinnamic acid in 50% acetonitrile containing 0.1% TFA.

### 2.3. MALDI-TOF/TOF-MS/MS Analysis and Protein Identification

The mass spectrometry (MS) spectra of digested peptides were performed on 5800 MALDI-TOF/TOF Plus mass spectrometer (Applied Biosystems, Foster City, CA, USA). The data were obtained in a positive MS reflector using a CalMix5 standard to adjust by an I 5800 TOF-TOF Proteomics Analyser (Applied Biosystems, Framingham, MA, USA). The databases used in this analysis were obtained from NCBI (http://www.ncbi.nlm.nih.gov; 6391 sequences) and SilkDB (http://www.silkdb.org/silkdb; 20361 sequences). The analysis of both the MS and MS/MS spectra data and the protein identification processes were implemented in accordance with a previous study [16]. The peptides of the quantified proteins were provided in Appendix A. Gene Ontology (GO) assignments were completed using Blast2GO (https://www.blast2go.com/) and obtained corresponding GO identification numbers (IDs) of the identified proteins using InterproScan 5.0 sequence search [17,18]. GO annotation results of the DEPs were provided in Appendix A. The Kyoto Encyclopedia of Genes and Genomes (KEGG) was used to perform pathway enrichment analysis of the identified proteins [19]. Descriptions of the KEGG pathways were provided in Appendix A.

### 2.4. Quantitative PCR

The genes selected according to the DEPs were investigated by quantitative PCR (qPCR) at the transcriptional level. Total RNA from the hemocytes and fat body of the infected and control samples was used to synthesize the first strand cDNA using a PrimeScript Reverse Transcriptase kit (TaKaRa, Dalian, China) according to the manufacturer’s instructions. Specific primers of genes for qPCR are listed in Appendix A. qPCR was performed as previously described [20]. The gene expression levels were calculated using the 2^−∆∆Ct^ method. There were three biological sample replicates, and each biological sample replicate included three independent experiments. The reference gene was *B*. *mori* ribosomal protein gene *BmRPL3*. The statistical analysis was conducted using ANOVA, followed by an LSD a posteriori test via SPSS statistical software (version 16.0; SPSS, Inc., Chicago, IL, USA).

## 3. Results

### 3.1. Identification of DEPs

In the interest of understanding the molecular mechanism of the late stage of *E. sorbillans*-parasitized silkworms at the protein level, we collected the hemolymph of female silkworm pupae on the first day of pupation and conducted 2-DE combined with MALDI-TOF/TOF-MS/MS analysis to identify the DEPs. The protein molecular mass varied within a normal range of approximately 10 to 116 kDa, and isoelectric point (pI) values ranged from 3 to 10 (Figure 1, Appendix A), suggesting that protein extraction was correctly performed and that most proteins from the hemolymph were obtained. A differentially expressed spot was defined as a spot with a 2.5-fold or greater change in intensity and frequency higher than 40%. According to the criteria, 36 upregulated and 14 downregulated spots were selected and subjected to MS/MS identification (Table 1). The DEPs ranged from 10 to 82 kDa in molecular mass and from pH 4 to 10 in pI. Detailed information, including the spot numbers, accession numbers, predicted molecular weights (MWs), pIs, sequence coverage, peptide count, fold change, and signal peptide, is provided in Table 1. The following eight innate immune proteins were upregulated in the infected group: PPO2 (spot 1), paralytic peptide binding protein (spot 4), antitrypsin (spot 6), CTL10 (spot 10), apolipophorin III (spot 22), translationally controlled tumor protein (spot 23), peptidoglycan recognition protein (spot 27), and ubiquitin-like protein SMT3 (spot 36) (Figure 1 and Table 1). Apoptosis-related proteins of cathepsin B (spot 43) and 14-3-3 zeta (spot 45) were significantly downregulated in the infected group (Figure 1 and Table 1). In this study, we analyzed the tissue expression patterns of the genes encoding the DEPs based on the microarray data. The raw data of 10 silkworm tissues on day 3 of the fifth instar were obtained from SilkMDB. We found many highly expressed genes in the epidermis amongst the 10 tissues (Appendix A). The expressed genes are defined as previously described [21]. This finding may be closely related to the epidermal invasion of maggots in the silkworm. Our data contribute to the understanding of the infection pathway of maggots.

### 3.2. GO Annotation and KEGG Pathway Enrichment Analysis of the DEPs

According to the GO annotation, 49 of the 50 identified DEPs were discovered with at least one GO match (Appendix A). Based on the GO database, the identified DEPs were classified into three categories: cellular component, molecular function, and biological process (Figure 2). In the cellular component category, the DEPs are involved in different cellular processes and the extracellular region of the cell and macromolecular complex were the main members. The DEPs are mainly related to binding and catalysis in the molecular function category. The biological process category showed DEPs that are mainly involved in cellular, metabolic, and single-organism processes. KEGG pathway enrichment analysis of the identified DEPs showed that the main pathways are involved in disease, energy metabolism, signaling pathways, and amino acid metabolism (Table 2 and Appendix A).

### 3.3. Verification of Gene Expressions by Quantitative PCR

To validate the 2-DE result, we also performed qPCR on some selected targets in the control and *E*. *sorbillans*-infected hemocytes and fat bodies. The insect hemolymph is an open circulatory system. During larval–pupal metamorphosis, the fat body maintains intracellular homeostasis and meets the requirements of metamorphosis. Thus, the fat body was also used to verify gene transcription of the DEPs in the hemolymph. The fat body and hemocytes were collected from day-1 female silkworm pupae at the same time point. The information on the five selected differentially expressed genes and *B. mori* ribosomal protein gene *BmRPL3* primers are presented in Appendix A. In the hemocytes and fat body, the transcriptional expression levels of phenoloxidase subunit 2 precursor (PPO2), CTL10, and peptidoglycan recognition protein (PGRP-S1) were increased with infection, whereas the gene expression levels of ecdysteroid-regulated 16 kDa protein (ESR16) and 14-3-3 protein zeta (14-3-3z) decreased with infection (Figure 3A,B). The induced fold-change of every gene was different, whereas the tendencies to expression changes were consistent between the hemocytes and fat body tissues. In summary, the changes in the gene transcription were consistent with their corresponding proteins in the 2-DE data.

## 4. Discussion

In this study, we successfully identified 50 host-responsive DEPs via MALDI-TOF/TOF-MS in the hemolymph of female silkworm pupae after *E*. *sorbillans* infection. Parasitoids induce host responses such as enhancing innate immunity proteins expression and cell apoptosis.

### 4.1. Innate Immune System Enhanced Resistance to E. sorbillans Infestation

Parasitoids induce host responses. In particular, the following eight innate immune proteins were successfully identified and upregulated in the infected group: PPO2 (spot 1), paralytic peptide binding protein (spot 4) [22,23], antitrypsin (spot 6) [24], CTL10 (spot 10), apolipophorin III (spot 22) [25], translationally controlled tumor protein (spot 23) [26,27], peptidoglycan recognition protein (spot 27) [28,29], and ubiquitin-like protein SMT3 (spot 36) [30,31]. Three proteins, PPO2, antitrypsin, and paralytic peptide binding protein, were identified as an immune adaptation against *E*. *bombycis* parasites in *B*. *mori* by Pradeep et al [8]. They all play important roles in innate immunity. Insect prophenoloxidase (PPO) is an important innate immunity protein [32]. The activation of PPO cleaving into active phenoloxidase (PO) by serine proteinase is required for a melanization cascade to isolate microorganisms from circulation and to then kill them [33]. Produced by hindgut cells, PPO is secreted into the hindgut content that induces the melanization of the hindgut content in silkworm [34]. BmCTL10 and BmMBP are the same C-type lectins (CTLs), and their amino acid sequence similarity is 99.7%; they participate in innate immune responses such as hemocyte nodule formation and PPO activation [4,35,36,37]. As a result of activation of the PPO cascade and nodule formation, the parasite is blackened in the host by the deposition of melanin and encapsulation. This indicates that the expressions of immune proteins are enhanced in the silkworm resistance of *E*. *sorbillans* parasite infestation.

### 4.2. Apoptosis Triggered in Response to E. sorbillans Infestation

Parasitic infection induces autophagy and cell apoptosis in insects [6]. Cathepsin B and 14-3-3 zeta both have important roles in apoptosis [38,39]. RNAi-mediated downregulation of cathepsin B or the absence of cathepsin B induces apoptosis in cancer [38,40,41]. Downregulation of cathepsin B can induce caspase-8-mediated apoptosis and initiates a partial extrinsic apoptotic cascade in SNB19 human glioma cells [40]. Strong 14-3-3 zeta protein expression acts in cell differentiation, proliferation, transformation, and prevention of apoptosis [42]. In particular, 14-3-3 protein zeta is a major regulator of apoptotic pathways in insects and vertebrates [43,44,45,46]; downregulation of 14-3-3 zeta sensitizes cells to apoptosis [39,47]. Cathepsin B (spots 43) and 14-3-3 zeta (spot 45) were successfully identified and were significantly downregulated in the hemolymph of parasitized female silkworm pupae. Cathepsin B was transcriptionally downregulated in the host *Manduca sexta* following wasp *Cotesia congregata* parasitism [48]. The downregulation of cathepsin B and 14-3-3 zeta may induce cell apoptosis in the hemolymph of *B*. *mori* following the invasion by the *E*. *sorbillans*. The larval epithelium of *B*. *mori* parasitized by *E*. *bombycis* showed cellular responses, such as signs of autophagy and apoptosis [8]. Enhanced expression of autophagy 5-like (*Atg5*), apoptosis-inducing Factor (*AIF*), and *caspase* genes coupled with the appearance of cell death symptoms indicate parasitism-induced activation of genetic machinery to modulate cell apoptosis in the epithelium [12]. Thus, cell apoptosis is triggered against the parasitism of *E*. *sorbillans* in *B*. *mori*.

Proteins related to growth and development, such as the ecdysteroid-regulated 16 kDa protein, were successfully identified. Ecdysteroid-regulated 16 kDa protein (ESR16) is triggered by the steroid hormone ecdysone at the onset of metamorphosis [49]. The developmentally regulated gene of ecdysteroid-regulated 16 kDa protein was downregulated in the hemolymph of *B*. *mori* after *E*. *sorbillans* infection, which indicates that the development of the host is affected to facilitate the growth of parasitic larvae of the *E*. *sorbillans* maggot.

## 5. Conclusions

In conclusion, we successfully identified 50 differentially expressed proteins (DEPs) in the hemolymph of the day-1 female silkworm pupae and their potential roles following silkworm maggot (*Exorista sorbillans*) parasitism using a proteomics-based approach. The expressions of immune proteins were enhanced, and cell apoptosis could be triggered against the parasitism of *E*. *sorbillans* in *B*. *mori*. To the best of our knowledge, this study is the first to report the identification of possible proteins from the *E*. *bombycis*-parasitized silkworms at the late stage of parasitism. Our findings expand the current knowledge on resistance in silkworm to *E*. *sorbillans* parasitization and provide a new perspective on the molecular mechanisms of dipteran parasitoid biology.

## Figures and Tables

**Figure 1 insects-10-00413-f001:**
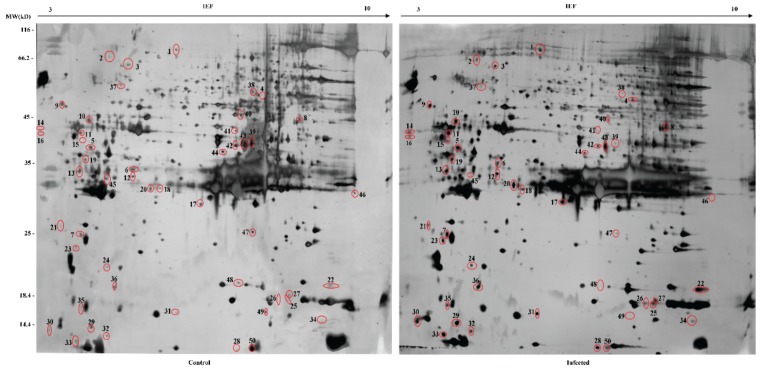
Two-dimensional gel electrophoresis (2-DE) maps of the control and infected hemolymph collected from day-1 female silkworm pupae: The differentially expressed proteins (DEPs) are indicated by circles and numeric labels, which correspond to the numbers presented in Table 1. All samples were processed in parallel.

**Figure 2 insects-10-00413-f002:**
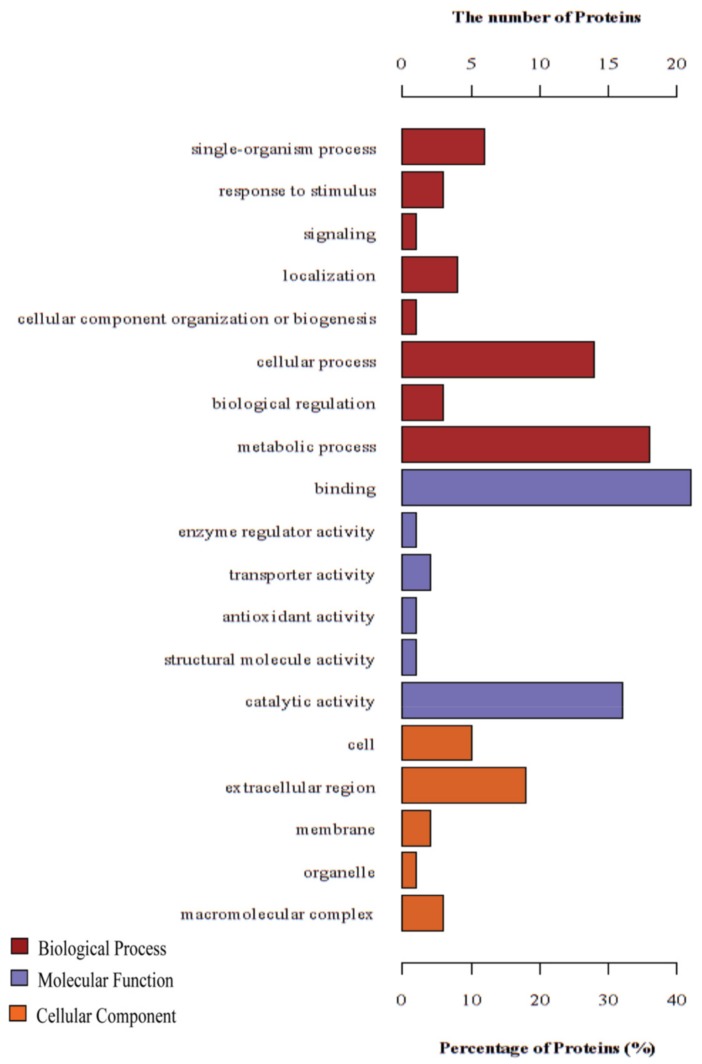
Distribution of the Gene Ontology (GO) terms for all proteins identified from the control and infected hemolymph: The DEPs were classified into cellular component, molecular function, and biological process by GO.

**Figure 3 insects-10-00413-f003:**
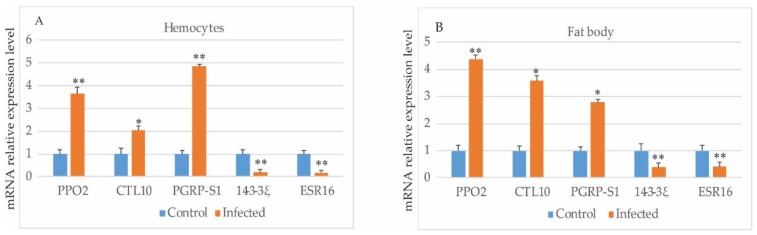
Expression profiles of five genes in the control and infected (**A**) hemocytes and (**B**) fat bodies collected from day-1 female silkworm pupae: For each gene, the transcriptional level of the control was set to 1. The data are the means ± SD of three independent experiments. Statistical analysis was performed using SPSS software. * *p* < 0.05, ** *p* < 0.01.

**Table 1 insects-10-00413-t001:** Data of the 50 DEPs identified by MALDI-TOF/TOF-MS/MS.

Spot No.	Protein Name	SilkDB Accession No.	NCBI Accession No.	Theoretical (kDa/pI)	Sequence Coverage(%)	Peptides Identified	Fold Change	Signal Peptide	Function
**Upregulated**									
1	phenoloxidase subunit 2 precursor	BGIBMGA013115-PA	gi|112983448	80.12/5.62	18	11	3.66	-	Serine protease involved in melanization
2	very low-density lipoprotein receptor isoform 1 precursor	BGIBMGA006214-PA	gi|160333138	61.54/5.20	4	3	4.53	-	Chitin metabolic
3	60 kDa heat shock protein, mitochondrial-like	BGIBMGA007349-PA	gi|512896628	61.06/5.51	22	8	2.61	-	Protein refolding
4	paralytic peptide binding protein	BGIBMGA010168-PA	gi|112983896	50.01/6.33	30	11	3.17	-	Extracellular region
5	uncharacterized protein LOC101739845	BGIBMGA002604-PA	gi|512931715	49.40/5.42	15	6	2.94	-	Farnesoic acid o-methyltransferase
6	antitrypsin	BGIBMGA009953-PA	gi|253809709	43.43/5.41	26	8	9.88	-	Peptidase inhibitor activity
7	failed axon connections isoform X1	BGIBMGA000552-PA	gi|512921311	42.89/5.31	11	3	3.64	-	N/A
8	proliferation-associated protein 2G4	BGIBMGA002493-PA	gi|512926720	42.14/7.13	33	9	3.24	-	Cellular process
9	DNA supercoiling factor	BGIBMGA001107-PA	gi|347326520	38.02/4.48	40	12	2.83	-	Calcium ion binding
10	C-type lectin 10	BGIBMGA006768-PA	gi|148298818	36.56/5.53	6	2	3.59	-	Carbohydrate binding
11	32 kDa apolipoprotein precursor	BGIBMGA002703-PA	gi|226501956	32.09/4.79	7	3	2.91	1	Pigment binding
12	spermidine synthase	BGIBMGA005897-PA	gi|512899761	32.43/5.54	26	8	3.15	-	Catalytic activity
13	small glutamine-rich tetratricopeptide repeat-containing protein alpha-like	BGIBMGA010000-PA	gi|827549620	31.43/4.88	8	2	5.77	-	Protein binding
14	uncharacterized protein LOC778506 isoform X1	—	gi|827559778	30.66/4.31	9	2	4.22	-	Cell surface glycoprotein
15	gasp precursor	BGIBMGA007677-PA	gi|114052326	29.09/4.82	7	1	24.35	1	Chitin binding
16	uncharacterized protein LOC778506 isoform X2	—	gi|827559780	28.45/4.25	9	2	4.15	1	N/A
17	low molecular mass 30 kDa lipoprotein 21G1 isoform X1	BGIBMGA004395-PA	gi|827538310	30.24/6.84	33	9	5.78	1	Extracellular region
18	low molecular 30 kDa lipoprotein PBMHPC-19-like precursor	BGIBMGA004398-PA	gi|525343846	28.50/5.72	13	3	5.77	1	Extracellular region
19	charged multivesicular body protein 5	BGIBMGA002470-PA	gi|512926615	25.26/4.71	16	2	4.34	-	Protein transport
20	Rab7	BGIBMGA007712-PA	gi|114051368	23.42/5.16	4	1	8.85	-	Small GTPase mediated signal transduction
21	uncharacterized protein LOC101746349	BGIBMGA006731-PA	gi|512923633	20.06/4.56	27	4	5.38	1	N/A
22	apolipophorin III	BGIBMGA013108-PA	gi|112983018	20.73/9.04	29	7	7.04	1	Defense response
23	translationallycontrolled tumor protein	BGIBMGA003073-PA	gi|112982880	19.86/4.66	13	2	31.59	-	Pathogen binding
24	translation initiation factor 5A	BGIBMGA007469-PA	gi|112982832	17.52/5.16	14	2	2.93	-	Translational frameshifting
25	abnormal wing disc-like protein	BGIBMGA007701-PA	gi|153791847	17.31/6.74	26	5	11.04	-	Nucleoside diphosphate phosphorylation
26	cyclophilin-like protein	BGIBMGA002429-PA	gi|60592747	17.96/7.74	17	2	3.81	-	Protein peptidyl-prolyl isomerization
27	peptidoglycan recognition protein	BGIBMGA008038-PA	gi|112983994	21.63/6.70	33	6	3.32	1	N-acetylmuramoyl-L-alanine amidase activity
28	probable pterin-4-alpha-carbinolamine dehydratase	—	gi|512899129	17.93/9.94	30	3	18.51	-	4-α-hydroxytetrahydrobiopterin dehydratase activity
29	odorant-binding protein 6 isoform X1	BGIBMGA008354-PA	gi|827551076	15.96/4.94	18	3	3.41	1	Odorant binding
30	E3 ubiquitin-protein ligase ZNRF2	BGIBMGA011980-PA	gi|512897556	21.91/5.42	7	1	4.56	-	Zinc ion binding
31	ribosomal protein S12	BGIBMGA004374-PA	gi|112982671	15.03/5.79	24	2	4.62	-	Structural constituent of ribosome
32	chemosensory protein 4 precursor	BGIBMGA004045-PA	gi|112983094	14.55/5.17	8	1	2.51	1	Transporters of pheromone/odor molecules
33	chemosensory protein 7 precursor	BGIBMGA004041-PA	gi|112983052	13.52/4.97	18	3	4.77	-	Transporters of pheromone/odor molecules
34	FK506-binding protein	BGIBMGA004331-PA	gi|114051243	11.82/7.85	43	4	10.66	-	Isomerase activity
35	uncharacterized protein LOC101736984 isoform X3	BGIBMGA007627-PA	gi|512916631	11.18/5.02	9	1	3.53	-	N/A
36	ubiquitin-like protein SMT3	BGIBMGA011581-PA	gi|112983974	10.31/5.29	23	3	**2.73**	-	Protein binding
**Downregulated**									
37	heat shock protein 83	BGIBMGA004612-PA	gi|112983556	82.42/4.98	3	2	0.11	-	Unfolded protein binding
38	ATP synthase	BGIBMGA001853-PA	gi|114052278	59.66/9.21	4	2	0.21	-	ATP binding
39	serine proteinase-like protein isoform X1	BGIBMGA009551-PA	gi|827563139	44.88/5.73	15	6	0.01	1	Serine-type endopeptidase activity
40	ornithine aminotransferase, mitochondrial	BGIBMGA003564-PA	gi|512922127	44.70/6.36	5	2	0.21	-	Pyridoxal phosphate binding
41	aldose 1-epimerase	BGIBMGA009232-PA	gi|512891308	39.71/5.88	27	7	0.28	-	Transaminase activity
42	aldo-ketoreductase AKR2E4-like isoform X1	BGIBMGA001348-PA	gi|512908850	38.99/5.82	10	5	0.26	1	Oxidoreductase activity
43	cathepsin B	BGIBMGA007061-PA	gi|112983908	37.56/5.95	18	5	0.17	-	Regulation of catalytic activity
44	aldose reductase-like isoform X1	BGIBMGA012152-PA	gi|512901366	35.84/6.10	20	6	0.39	-	Oxidoreductase activity
45	14-3-3 protein zeta	BGIBMGA002644-PA	gi|114050901	28.17/4.90	6	2	0.21	-	Protein domain specific binding
46	vacuolar ATP synthase subunit E	BGIBMGA010247-PA	gi|114052088	26.12/8.98	17	3	0.19	-	ATP hydrolysis
47	tyrosine-protein phosphatase Lar	BGIBMGA012106-PA	gi|512933991	22.75/6.29	7	1	0.18	1	Protein binding
48	diapause bioclock protein	BGIBMGA002907-PA	gi|68144076	18.29/6.12	27	3	0.15	1	Superoxide dismutase activity
49	ecdysteroid-regulated 16 kDa protein precursor	BGIBMGA008405-PA	gi|151301100	15.82/5.92	12	1	0.01	1	Ecdysteroid level regulated
50	chemosensory protein 5 precursor	BGIBMGA004065-PA	gi|112983054	14.26/6.89	22	3	0.23	1	RNA-binding

Note: - no signal peptide was predicted and no SilkDB accession number was found; N/A not applicable.

**Table 2 insects-10-00413-t002:** Kyoto Encyclopedia of Genes and Genomes (KEGG) pathway enrichment analysis of DEPs.

Number	Pathway	Pathway ID	Accession No.	Description	Fold Change
1	Galactose metabolism	ko00052	gi|512891308	aldose 1-epimerase	0.28
			gi|512908850	aldehyde reductase	0.26
			gi|512901366	aldehyde reductase	0.39
2	Huntington’s disease	ko05016	gi|60592747	peptidyl-prolyl isomerase	3.81
			gi|114052278	ATPase	0.21
			gi|68144076	superoxide dismutase	0.15
3	Glycerolipid metabolism	ko00561	gi|512908850	aldehyde reductase	0.26
			gi|512901366	aldehyde reductase	0.39
4	Longevity regulating pathway—worm	ko04212	gi|512896628	chaperonin	2.61
			gi|114050901	protein binding	0.21
5	Lysosome	ko04142	gi|112983908	cathepsin B	0.17
			gi|151301100	Niemann–Pick C2 protein	0.01
6	Pentose and glucuronate interconversions	ko00040	gi|512908850	aldehyde reductase	0.26
			gi|512901366	aldehyde reductase	0.39
7	Fructose and mannose metabolism	ko00051	gi|512908850	aldehyde reductase	0.26
			gi|512901366	aldehyde reductase	0.39
8	Parkinson’s disease	ko05012	gi|60592747	peptidyl-prolyl isomerase	3.81
			gi|114052278	ATPase	0.21
9	Amoebiasis	ko05146	gi|253809709	serpin B	9.88
			gi|114051368	Ras-related protein	8.85
10	Oxidative phosphorylation	ko00190	gi|114052278	ATPase	0.21
			gi|114052088	ATPase	0.19
11	Tuberculosis	ko05152	gi|512896628	chaperonin	2.61
			gi|114051368	Ras-related protein	8.85
12	Antigen processing and presentation	ko04612	gi|112983556	molecular chaperone	0.11
			gi|112983908	cathepsin B	0.17
13	Phagosome	ko04145	gi|114051368	Ras-related protein	8.85
			gi|114052088	ATPase	0.19
14	Endocytosis	ko04144	gi|512926615	charged multivesicular body protein	4.34
			gi|114051368	Ras-related protein	8.85
15	PI3K-Akt signaling pathway	ko04151	gi|112983556	molecular chaperone	0.11
			gi|114050901	protein binding	0.21
16	Arginine and proline metabolism	ko00330	gi|512899761	spermidine synthase	3.15
			gi|512922127	ornithine–oxo-acid transaminase	0.21
17	NOD-like receptor signaling pathway	ko04621	gi|112983556	molecular chaperone	0.11
18	Legionellosis	ko05134	gi|512896628	chaperonin	2.61
19	Epstein-Barr virus infection	ko05169	gi|114050901	protein binding	0.21
20	Glycolysis/Gluconeogenesis	ko00010	gi|512891308	aldose 1-epimerase	0.28

Note: phosphatidylinositol-3-kinase (PI3K); phosphorylation of protein kinase B (Akt); nucleotide-binding and oligomerization domain (NOD).

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
