# Peer review of "Comparative Proteomic Analysis Reveals Immune Competence in Hemolymph of Bombyx mori Pupa Parasitized by Silkworm Maggot Exorista sorbillans"

_insects, 2019, doi:10.3390/insects10110413_

Round 1

Reviewer 1 Report

Dear Author/s,

Re-submitted manuscript “Comparative Proteomic Analysis Reveals Response Mechanistic Insights into Hemolymph of Bombyx mori Parasitized by Silkworm maggot, Exorista sorbillans” still have lot of scientific errors and extensively jam packed with language mistakes. As I felt before that you (author/s) have done lots of work but unable to justify and explain the results appropriately, hence it is highly advisable rewrite the results what you found from various analysis. Place the results only in tables and figure format (including supplementary) and just indicate their address is not enough. Why did you analyze, what you want to observe, what is major objective of the study, all these points are missing beginning from introduction.

At present, correction at every point is impossible for me. Here, I have highlighted some of the lines in the attached manuscript (Please find the attachment) and broadly, suggesting some broad corrections for your referral. Follow that and similarly do the other amendments and proofreading accordingly. Apart from that, I would suggest to take the help of technical support of journal’s English editing services or any known expert of English language.

Title of the manuscript is become vague now. 2DE were done in hemolymph whereas qPCR were done in fat body and in abstract and results you mentioned (Of these proteins, the expression patterns of genes encoding the five DEPs according to quantitative PCR analyses were consistent with the two-dimensional gel electrophoresis results). Unwanted, Irrelevant, Surplus (26-33), references have cited in the paper which is highlighted over there. Programmed cell death is called as Apoptosis. Both words together in heading misleading the discussion. Repetitions of lines and facts in introduction and discussion such as “Infection of B. mori larvae by E. sorbillans induces host responses” Figure 3- You are presenting the values as increasing and decreasing compare it with control. Is it significant change or the due the height of bar you have written so. SD/SEM and asterisks are missing. In methodology, Line No. 76 “The normal larvae were used as a control.” While the rest other experiment were done in Pupae, is it scientifically correct to compare their proteomic expression.

There are still numerous corrections are demanded in the re-submitted manuscript for which author/s should take the expert opinion of the respective field. Please, find the attachments for some additional amendments in yellow highlights in the manuscript.

All the best.

Author Response

Response to Reviewer 1 Comments Point 1: Re-submitted manuscript “Comparative Proteomic Analysis Reveals Response Mechanistic Insights into Hemolymph of Bombyx mori Parasitized by Silkworm maggot, Exorista sorbillans” still have lot of scientific errors and extensively jam packed with language mistakes. As I felt before that you (author/s) have done lots of work but unable to justify and explain the results appropriately, hence it is highly advisable rewrite the results what you found from various analysis. Place the results only in tables and figure format (including supplementary) and just indicate their address is not enough. Why did you analyze, what you want to observe, what is major objective of the study, all these points are missing beginning from introduction. Response 1: Thanks for the reviewer’s good evaluation and kind suggestion. Due to your suggestion, (ⅰ): we carefully revised our manuscript including sentence smoothness, punctuation and spelling, etc, and our manuscript has been revised in MDPI for English editing by selecting specialist editing (English editing ID: English-12859); (ⅱ): the revised manuscript has also been supplemented, adjusted and improved. The revisions are highlighted using the "Red Font" and "Track Changes" function in Microsoft Word. Point 2: At present, correction at every point is impossible for me. Here, I have highlighted some of the lines in the attached manuscript (Please find the attachment) and broadly, suggesting some broad corrections for your referral. Follow that and similarly do the other amendments and proofreading accordingly. Apart from that, I would suggest to take the help of technical support of journal’s English editing services or any known expert of English language. Response 2: Thanks for the reviewer’s good evaluation and kind suggestion. We carefully revised each part of our manuscript, especially, which you had pointed out. We also sent the manuscript to MDPI for English revision. The revisions are highlighted using the "Red Font" and "Track Changes" function in Microsoft Word. The advices from you have provided with great helps to us in the current work. Thank you very much. Point 3: Title of the manuscript is become vague now. Response 3: Thanks for the reviewer’s kind suggestion. We are not accurate enough to grasp the title of our manuscript. According to the main results we obtained, we changed the title into "Comparative Proteomic Analysis Reveals Immune Competence in Hemolymph of Bombyx mori Pupa Parasitized by Silkworm maggot, Exorista sorbillans" replacing the previous "Comparative Proteomic Analysis Reveals Response Mechanistic Insights into Hemolymph of Bombyx mori Parasitized by Silkworm maggot, Exorista sorbillans". Point 4: 2DE were done in hemolymph whereas qPCR were done in fat body and in abstract and results you mentioned (Of these proteins, the expression patterns of genes encoding the five DEPs according to quantitative PCR analyses were consistent with the two-dimensional gel electrophoresis results). Response 4: Thanks for the reviewer’s good evaluation. Silkworm hemolymph is an open circulatory system. Fat body is the key one of the specific tissues that produce and release proteins into the hemolymph. Otherwise, during larval–pupal metamorphosis, the fat body maintains intracellular homeostasis and meets the requirements of metamorphosis. The fat body and hemolymph were collected from female silkworm pupae on the first day of pupation. This time is the terminal stage of silkworm larval–pupal metamorphosis. The fat body and hemolymph were collected at the same time-point. Thus, we used fat body to verify gene transcription of differentially expressed proteins (DEPs) in hemolymph. In the previous manuscript, we didn't elaborate enough on this background. We are sorry. In the revision, new supplemental contents are prepared and can be found in Line 94-96, page 3, and Line 187-193, page 12. Point 5: Unwanted, Irrelevant, Surplus (26-33), references have cited in the paper which is highlighted over there. Response 5: Thanks for the reviewer’s kind suggestion. We have removed the redundant references in the revision. The revisions are highlighted using the "Track Changes" function in Microsoft Word. Point 6: Programmed cell death is called as Apoptosis. Both words together in heading misleading the discussion. Response 6: Thanks for the reviewer’s kind suggestion. We have made corrections according to your advice. Point 7: Repetitions of lines and facts in introduction and discussion such as “Infection of B. mori larvae by E. sorbillans induces host responses”. Response 7: Thanks for the reviewer’s good evaluation and kind suggestion. After carefully checking the manuscript, we found that the description of "Infection of B. mori larvae by E. sorbillans induces host responses" is reduplicative. In the revision, we used "pupae parasitized by E. sorbillans", "following the invasion by E. sorbillans", and "E. sorbillans-parasitized silkworm pupae" to replace "Infection of B. mori larvae by E. sorbillans induces host responses" In the corresponding positions, respectively. Additional, the description of "Infection of B. mori larvae" is vague. The day 2 fifth instar larvae were exposed to the mated gravid females of E. sorbillans for oviposition. After about two days, the eggs are hatched and the maggots invade the day 4 fifth instar silkworm larvae body. The experimental samples of hemolymph and fat body were collected from the infected and control female silkworm pupae on the first day of pupation. The description of scientific fact is "Infection of B. mori pupae" in the revision. Thank you very much for your careful reviews in our work. Point 8: Figure 3- You are presenting the values as increasing and decreasing compare it with control. Is it significant change or the due the height of bar you have written so. SD/SEM and asterisks are missing. Response 8: Thanks for the reviewer’s good evaluation. The related content has been revised. Point 9: In methodology, Line No. 76 “The normal larvae were used as a control.” While the rest other experiment were done in Pupae, is it scientifically correct to compare their proteomic expression. Response 9: Thanks for the reviewer’s good evaluation. The description of “The normal larvae were used as a control” was not accurate in the previous manuscript. We are sorry. The description of scientific fact is "Control larvae were kept without infestation" in the revision. The day 2 fifth instar larvae were exposed to the mated gravid females of E. sorbillans for oviposition. After about two days, the eggs are hatched and the maggots invade the day 4 fifth instar silkworm larvae body. The experimental samples of hemolymph and fat body were collected from the infected and control female silkworm pupae on the first day of pupation. In the revision, new supplemental contents are prepared and can be found in Line 84-89, page 2. Thank you very much for your careful reviews in our work. Point 10: There are still numerous corrections are demanded in the re-submitted manuscript for which author/s should take the expert opinion of the respective field. Please, find the attachments for some additional amendments in yellow highlights in the manuscript. Response 10: Thanks for the reviewer’s good evaluation and kind suggestion. According to your suggestion, we have made the adjustment in the revision. The revisions are highlighted using the "Red Font" and "Track Changes" function in Microsoft Word. The advices from you have provided with great helps to us in the current work and will improve our level of scientific research in the future work. Thank you very much.

Reviewer 2 Report

The current revised MS by Xu et al. should address the remaining issues raised by this reviewer to be acceptable in Insects.

1) Line 38-50/200: Apparent changes in font size.
2) The whole introduction should be re-organized with clear logic. The sequence of 2nd and 3rd paragraph is not well defined.
3) Fig. 2: The aspect ratio should be kept when zooming in the figure.
4) Fig. 3: It should be PPO2 instead of PP02. How about the changes in the mRNA level in hemocytes? Western blot results are still required
5) Figure legends should be explained in a more detailed manner.
6) If the authors are considering the apoptosis pathway as one of the major regulated pathways, more evidence and experiments are expected.

Author Response

Response to Reviewer 2 Comments Point 1: Line 38-50/200: Apparent changes in font size. Response 1: Thanks for the reviewer’s good evaluation. The font has been adjusted according to the requirements of the journal. Thank you very much for your careful reviews in our work. Point 2: The whole introduction should be re-organized with clear logic. The sequence of 2nd and 3rd paragraph is not well defined. Response 2: Thanks for the reviewer’s good evaluation and kind suggestion. We carefully revised each part of our manuscript, especially, which you had pointed out. We had re-organized "Introduction" in order to describe our study with a clear logic. The changes in "Introduction" can be found in Line 43-72, page 2. The revisions in other parts are highlighted using the "Red Font" and "Track Changes" function in Microsoft Word. The advices from you have provided with great helps to us in the current work. Thank you very much. Point 3: Fig. 2: The aspect ratio should be kept when zooming in the figure. Response 3: Thanks for the reviewer’s kind suggestion. In the revision, the aspect ratio of Figure 2 can be kept when zooming in the figure. Thank you very much for your careful reviews in our work. Point 4: Fig. 3: It should be PPO2 instead of PP02. How about the changes in the mRNA level in hemocytes? Western blot results are still required. Response 4: Thanks for the reviewer’s good evaluation. In the revision, we used PPO2 to replace PPo2 in Figure 3. Silkworm hemolymph is an open circulatory system. Fat body is the key one of the specific tissues that produce and release proteins into the hemolymph. Otherwise, during larval–pupal metamorphosis, the fat body maintains intracellular homeostasis and meets the requirements of metamorphosis. The fat body and hemolymph were collected from female silkworm pupae on the first day of pupation. This time is the terminal stage of silkworm larval–pupal metamorphosis. The fat body and hemolymph were collected at the same time-point. Thus, we used fat body to verify gene transcription of differentially expressed proteins (DEPs) in hemolymph. In the previous manuscript, we didn't elaborate enough on this background. We are sorry. In the revision, new supplemental contents are prepared and can be found in Line 94-96, page 3, and Line 187-193, page 12. Additional, the day 2 fifth instar larvae were exposed to the mated gravid females of E. sorbillans for oviposition. After about two days, the eggs are hatched and the maggots invade the day 4 fifth instar silkworm larvae body. The parasitoid maggot completes larval stages inside the silkworm fifth instar larvae for about 5 days. The experimental samples of hemolymph and fat body were collected from the infected and control female silkworm pupae on the first day of pupation. This time was the late stage and the third (final) instar larva of endoparasitic maggot. Then the third instar maggot elicited from the silkworm body and pupated outside. In the late stage, in order to endure parasitism, the host silkworm shown long-playing and continuous requisite of the oxidative stress reaction and anti-oxidative activity in hemocytes. Before the third instar maggot eliciting from the silkworm body, the hemocytes of silkworm were badly damaged. Thus, we didn't select the hemocytes for the study. Each sample of hemolymph was collected from about 60 female silkworm pupae. Three replicates were performed. We strictly controlled the experimental processes. The processes of samples handling, isoelectric focusing (IEF), two-dimensional gel electrophoresis (2-DE), electrophoresis gel washing, fixing and staining, and spots scanning and analyzing were under the same conditions. Meanwhile, the fold changes of ≥2.5 or ≤0.4 were used to identify differently expressed protein spots. Moreover, two-dimensional electrophoresis (2-DE) has an obvious strength in all proteomics technologies that the separated protein spot was a single protein, within isoelectric focusing based on isoelectric point of proteins, and electrophoresis based on molecular weight of proteins. Hence, the differentially expressed proteins (DEPs) were identified with 2-DE technology and combined with some of them transcriptional results, which this study method is feasible. Undeniably, it is perfect in combining with the western blot results. The advices from you have provided with great helps to us in the current work and will improve our level of scientific research in the future work. Thank you very much. Point 5: Figure legends should be explained in a more detailed manner. Response 5: Thanks for the reviewer’s kind suggestion. We have added the details in the revision. The revisions are highlighted using the "Track Changes" function in Microsoft Word. Point 6: If the authors are considering the apoptosis pathway as one of the major regulated pathways, more evidence and experiments are expected. Response 6: Thanks for the reviewer’s kind suggestion. Parasitism-induced apoptosis in the epidermis of host silkworm in the early stage that has been reported in the previous studies (references 8, 9, and 12 in the revision), which indicates parasitism-induced activation of apoptosis in the host silkworm. Our study found the parasitism-induced apoptosis in hemolymph in the late stage. Cathepsin B and 14-3-3 zeta are evolutionarily conserved proteins through protein sequence blasting in the non-redundant protein sequences (nr) database in NCBI. Cathepsin B and 14-3-3 zeta both have important roles in apoptosis. Down-regulations of cathepsin B and 14-3-3 zeta induce apoptosis that has been reported in the previous researches. In our study, proteins of cathepsin B and 14-3-3 zeta were significantly down-regulated in E. sorbillans-parasitized silkworm pupae. Thus, apoptosis is triggered against the parasitism of E. sorbillans in B. mori. Thank you very much for your careful reviews in our work.

Round 2

Reviewer 1 Report

Dear Author,

The present form of the paper is appears upgraded now. Although, I will again advice to the authors that recheck/revise the paper scientifically (read thoroughly and carefully) one more time at your end and minimize the mistakes as much as you can, such as mentioned below.

Figure No. 3- The significant values (asterisk sign on the bar graph) are everywhere on (control and infected) what does it mean? Its look humorous and witty that PPO2 is more than 4-fold induced; which is least significant (P<0.05) while, PGRP-1 and CTL10 are only 2-fold and 3-fold induced (P<0.01) respectively and they are more significant. How come it is possible, it shows without calculating the values; authors has just put the asterisk sign on every bars, to fulfill the author comments. Correct the mistakes judiciously. Table-1 needs intensive formatting; subheading in bold as you are dividing the table into two parts viz. up-regulating and down-regulating genes. Validate all the minor mistakes attentively throughout the manuscript at your level.

All the best for your upcoming research.

Author Response

Response to Reviewer 1 Comments

Point 1: The present form of the paper is upgraded now. Although, I will again advice to the authors that recheck/revise the paper scientifically (read thoroughly and carefully) one more time at your end and minimize the mistakes as much as you can, such as mentioned below.

Response 1: Thanks for the reviewer’s good evaluation and kind suggestion. Due to your suggestion, we carefully revised our manuscript, especially, which you had pointed out. The amendments including some mistakes, font form, and sentence smoothness have been made in the revised version. The revisions are highlighted using the "Red Font" and "Track Changes" function in Microsoft Word. Thank you very much for your careful reviews in our work.

Point 2: Figure No. 3- The significant values (asterisk sign on the bar graph) are everywhere on (control and infected) what does it mean? Its look humorous and witty that PPO2 is more than 4-fold induced; which is least significant (P<0.05) while, PGRP-1 and CTL10 are only 2-fold and 3-fold induced (P<0.01) respectively and they are more significant. How come it is possible, it shows without calculating the values; authors has just put the asterisk sign on every bars, to fulfill the author comments. Correct the mistakes judiciously.

Response 2: Thanks for the reviewer’s good evaluation and kind suggestion. We are so sorry. We made mistakes in our manuscript. Relative expression levels were calculated using the 2-∆∆Ct method. Statistical analysis was performed using SPSS software (used in the revised version). We carefully revised our manuscript. The careful and precise reviews from you have provided with great helps to us in the current work. Thank you very much.

Point 3: Table-1 needs intensive formatting; subheading in bold as you are dividing the table into two parts viz. up-regulating and down-regulating genes. Validate all the minor mistakes attentively throughout the manuscript at your level.

Response 3: Thanks for the reviewer’s kind suggestion. We have revised one by one according to your suggestions. Thank you very much for your careful reviews in our work.

Reviewer 2 Report

The revised manuscript by Xu et al. addressed some of the issues raised by this reviewer. However, there are still missing points that are required to be resolved in the following round of the revision.

1) The authors argued that fat body tissues are the major immuno responders against stimuli. However, it still needs evidence to support the logic of why they investigate the hemolymph proteins instead of the proteins from the fat body. The qPCR results from hemocytes seem to be essential. Moreover, it is known that many hemolymph proteins are secreted from fat bodies and/or another tissue cell, for example, hemocytes. Some of the proteins from this study are cyctosol proteins. Some explanations and discussions are expected.

2) Figure 3: why the authors marked * everywhere? All of the authors should proofread the whole MS to make sure there is no such kind of mistakes.

Author Response

Response to Reviewer 2 Comments

Point 1: The revised manuscript by Xu et al. addressed some of the issues raised by this reviewer. However, there are still missing points that are required to be resolved in the following round of the revision.

Response 1: Thanks for the reviewer’s good evaluation and kind suggestion. Due to your suggestion, (ⅰ): we carefully revised our manuscript, especially, which you had pointed out. The amendments including some mistakes, font form, and sentence smoothness have been made in the revised version; (ⅱ): the qPCR results from hemocytes have been supplemented in the revised manuscript with the corresponding content. The revisions are highlighted using the "Red Font" and "Track Changes" function in Microsoft Word. Thank you very much for your careful reviews in our work.

Point 2: The authors argued that fat body tissues are the major immuno responders against stimuli. However, it still needs evidence to support the logic of why they investigate the hemolymph proteins instead of the proteins from the fat body. The qPCR results from hemocytes seem to be essential. Moreover, it is known that many hemolymph proteins are secreted from fat bodies and/or another tissue cell, for example, hemocytes. Some of the proteins from this study are cyctosol proteins. Some explanations and discussions are expected.

Response 2: Thanks for the reviewer’s good evaluation and kind suggestion. The careful and precise reviews, and the strategic and constructive advices from you have provided with great helps to us in the current work. In our previous understanding and view, the endoparasitic maggot is the third (final) instar larva in the late stage, the larva becomes larger and is more active in feeding during this period. It is speculated that the host silkworm is also more active in responding to parasitism. The silkwormsʹ hemolymph is an open circulatory system, the tissue of fat body is the primary site of immune response and occupies a large proportion in day 1 female silkworm pupae. Thus we select to identify the possible proteins in hemolymph and verify gene expression in fat body from the E. bombycis-parasitized silkworms at the late stage of parasitism. Due to your strategic and constructive advices, we realize that we didn't know enough before. Exogenous and not-self endoparasitic maggot is always in parasitism. To combat infection, the silkworm relies on multiple innate defense reactions. The fat body produces humoral response molecules including antimicrobial peptides (non-contacting and signaling transduction). The hemocytes participate in phagocytosis and encapsulation of foreign intruders in the hemolymph (direct foreign intruders contacting components). Many of the responses may be shared in the two major immune tissues. The qPCR results from hemocytes have supplemented in the revised manuscript. The revisions are highlighted using the "Red Font" and "Track Changes" function in Microsoft Word. The advices from you have provided with great helps to us in the current work. In our future work on silkworm parasitized by maggots, we have obtained the inspirations and ideas from your strategic and constructive suggestions. Thank you very much.

Point 3: Figure 3: why the authors marked * everywhere? All of the authors should proofread the whole MS to make sure there is no such kind of mistakes.

Response 3: Thanks for the reviewer’s kind suggestion. We are so sorry. We made mistakes in our manuscript. We carefully revised our manuscript. Thank you very much for your careful reviews in our work.

Round 3

Reviewer 2 Report

This reviewer has no further comments.

Author Response

Point 1: English language and style are fine/minor spell check required.

Response 1: Thanks for the reviewer’s good evaluation and kind suggestion. We have revised the manuscript according to your kind advices. Our manuscript has been edited by MDPI for English editing again (English editing ID: English-13987). The new revisions are highlighted using the "Light blue Font" and "Track Changes" function in Microsoft Word. The gene expression levels were calculated using the 2-∆∆Ct method. There were three biological sample replicates, and each biological sample replicate included three independent experiments. The reference gene was B. mori ribosomal protein gene BmRPL3. The statistical analysis was conducted using ANOVA, followed by an LSD a posteriori test via SPSS statistical software (version 16.0; SPSS, Inc., Chicago, IL, USA). The information on the five selected differentially expressed genes and B. mori ribosomal protein gene BmRPL3 primers are presented in Table S4. These supplementary contents can be found in Line 140-146 (page 4) in "Materials and Methods" section in the revision. Thank you very much for your careful reviews in our work.

This manuscript is a resubmission of an earlier submission. The following is a list of the peer review reports and author responses from that submission.

Round 1

Reviewer 1 Report

Dear Author/s,

Submitted manuscript “Comparative proteomic analysis of hemolymph indicating immune response activated in silkworm parasitized by silkworm maggot, Exorista sorbillans” is extensively jam packed with erroneous grammar and language mistakes. I feel that you (author/s) have done lots of work but unable to justify and explain the results appropriately.

Here, I am suggesting some of the broad corrections to make the report concise and meaningful, apart from the language correction. For that (English language), I would suggest to take the help of technical support of journal’s English editing services or any known expert of English language.

Major issues: Why did the author has analyzed the microarray data from SilkMDB and unnecessary trying to correlate with the results of particular genes of various tissues of silkworm, which belong to different experiment setup and their work, was done on hemocytes. Whereas, your work is confined only to the hemolymph protein and with Exorista sorbillans parasitic infection. Justify how figure 4 is actually belongs to the objective of your work firstly.

Title of the paper stated “comparative proteomic analysis of hemolymph indicating immune response activated in silkworm” To fulfill that authors have analyzed the 50 Differentially Expressed Protein (DEP) by means of 2-DE and MALDI-TOF, is it necessary to explain the role of every proteins, which are not playing any role in Immunity?

Not even this, in discussion section also lot of discussion is belongs to cellular and molecular processes with irrelevant literature. Precisely, Authors have to decide the direction of the manuscript and significantly present the results and discussion in that direction only. Or else, change the title of the paper and present the results under 2 or 3 subheadings accordingly viz. role immunity, cellular compartment or molecular role if any.

Overall, Authors have to work more on paper writing, (including language) and primarily arrange the results more captive way. Hence, it is advisable that analyze the results and revise the paper one more time accordingly.

All the best.

Author Response

Response to Reviewer 1 Comments

Point 1: Here, I am suggesting some of the broad corrections to make the report concise and meaningful, apart from the language correction. For that (English language), I would suggest to take the help of technical support of journal’s English editing services or any known expert of English language.

Response 1: Thanks for the reviewer’s good evaluation and kind suggestion. We are not native English speakers. It may not be concise and interesting in the grammar, wording and other aspects of English. Therefore, the manuscript has been embellished and revised by proof-reading-service (https://www.proof-reading-service.com/en/) before the submission.

Point 2: Why did the author has analyzed the microarray data from SilkMDB and unnecessary trying to correlate with the results of particular genes of various tissues of silkworm, which belong to different experiment setup and their work, was done on hemocytes. Whereas, your work is confined only to the hemolymph protein and with Exorista sorbillans parasitic infection. Justify how figure 4 is actually belongs to the objective of your work firstly.

Response 2: Thanks for the reviewer’s kind suggestion. According to his/her advices, we had added relevant content, the revised details can be found in Line 166-171, page 2. Day 3 of the fifth instar of silkworm is the boundary for whole larval development stage. The fifth instar of silkworm feeds and grows quickly before this time point, most biological processes may be similar during successive feeding stages at and before this time point. Thus, learning the tissue expressional preferences via investigating the spatial expression profile of genes encoding the identified proteins in hemolymph at this time point, it will be helpful to expand and enrich biological functions of the DEPs.

Point 3: Title of the paper stated “comparative proteomic analysis of hemolymph indicating immune response activated in silkworm” To fulfill that authors have analyzed the 50 Differentially Expressed Protein (DEP) by means of 2-DE and MALDI-TOF, is it necessary to explain the role of every proteins, which are not playing any role in Immunity? Not even this, in discussion section also lot of discussion is belongs to cellular and molecular processes with irrelevant literature. Precisely, Authors have to decide the direction of the manuscript and significantly present the results and discussion in that direction only. Or else, change the title of the paper and present the results under 2 or 3 subheadings accordingly viz. role immunity, cellular compartment or molecular role if any. Overall, Authors have to work more on paper writing, (including language) and primarily arrange the results more captive way. Hence, it is advisable that analyze the results and revise the paper one more time accordingly.

Response 3: Thanks for the reviewer’s good evaluation and kind suggestion. It's very important. Due to your suggestion, I found some shortcomings in my current work. We had made revisions and adjustments in the parts of introduction (in Line 40-58) results (in Line 166-171) and discussion (in Line 208-212, and 240-245). We had added 2 subheadings (in Line 207, and 246). I will improve the level of scientific research and achieve more results according to your suggestion in the future work.

Reviewer 2 Report

This manuscript by Xu et al. described the comparative proteomic analysis of hemolymph proteins in silkworms invaded by a kind of maggot, Exorista sorbillans. Based on their 2-DE gel, 50 DEPs were identified and some of which were verified by qPCR. This MS might be interesting to the researchers who are focusing on the maggot or silkworm topics. However, the authors present their results in a very bad and confusing manner.

1) The whole MS should have been checked by a native English speaker.

2) The significance of the study should be emphasized by adding more introduction and discussion. The reviewer also feels that this area might be too narrow and will be not very interesting to the readers.

3) A pathway map should be very helpful to understand which genes are up/down-regulated. This kind of data presentation is very essential when for example, RNA-seq or 2-DE is studied.

4) The authors should employ some antibodies to verify some of their DEPs. Although the mRNA expression could, in some cases, reflect the protein expressions, other approaches should be also utilized to validate the results.

5) The discussion section is quite loose. The authors should give a focus in each paragraph. 

Author Response

Response to Reviewer 2 Comments

Point 1: The whole MS should have been checked by a native English speaker.

Response 1: Thanks for the reviewer’s kind advice. We are not native English speakers. It may not be concise and interesting in the grammar, wording and other aspects of English. Therefore, the manuscript has been embellished and revised by proof-reading-service (https://www.proof-reading-service.com/en/) before the submission.

Point 2: The significance of the study should be emphasized by adding more introduction and discussion. The reviewer also feels that this area might be too narrow and will be not very interesting to the readers.

Response 2: Thanks for the reviewer’s kind suggestion. According to his/her advices, we had added relevant content, the revised details can be found in introduction (in Line 40-58) and discussion (in Line 208-212, and 240-245). Silkworm, Bombyx mori are the the economically important silk-producing lepidopteran insects and commercially reared in many countries such as China, India, Japan, Brazil, Uzbekistan, etc.

Point 3: A pathway map should be very helpful to understand which genes are up/down-regulated. This kind of data presentation is very essential when for example, RNA-seq or 2-DE is studied

Response 3: Thanks for the reviewer’s good evaluation and kind suggestion. Due to your suggestion, I found some shortcomings in my current work. Fifty differentially expressed proteins (DEPs) were identified, thirty-six proteins were up-regulated, and fourteen proteins were down-regulated. Moreover, we also described their functions in Table 1. After your advice, I will improve the skills of presenting the results in future work.

Point 4: The authors should employ some antibodies to verify some of their DEPs. Although the mRNA expression could, in some cases, reflect the protein expressions, other approaches should be also utilized to validate the results.

Response 4: Thanks for the reviewer’s good evaluation and kind suggestion. It's very important. Due to your suggestion, I found some shortcomings in my current work. I will improve the level of scientific research and achieve more results according to your suggestion in the future work.

Point 5: The discussion section is quite loose. The authors should give a focus in each paragraph.

Response 5: Thanks for the reviewer’s kind suggestion. According to his/her advices, we had added 2 subheadings in discussion section (in Line 207, and 246). We also had made revisions and adjustments in the parts of introduction (in Line 40-58) results (in Line 166-171) and discussion (in Line 208-212, and 240-245).

Round 2

Reviewer 1 Report

Dear Authors,

I found that paper has not improvised well at many fronts and not been properly answered the referees comments. As you responded, “I will improve the level of scientific research and achieve more results according to your suggestion in the future work”. This statement is neither the satisfactory answer of reviewer queries nor the explanation asked for the present work. Whatever the addition you have made in the paper is not sufficient. The response to the queries are illogical and not enough worthy for a research paper going to publish among the scientific community. Hence, please do the needful efforts to make the paper better, as you have done good experiments, which should be appreciated.

I would suggest please, look into my earlier remarks and try to take out/filter the data according to the comments including language and formatting.

All the very best.

Reviewer 2 Report

The revised MS by Xu et al. resolved some of the comments from this reviewer. However, the current revised MS should undergo another round revision to address the remaining issues raised by this reviewer. More effects must be done to improve this MS significantly to meet the requirement of Insects journal.

1) Style. This reviewer feels astonished that even the font size of author names are not consistent. The authors should pay more attention to their MS.

2) Regarding "Comment #3" & "Comment #4", the authors should do some summary and experiments to improve the level of their manuscript. It makes no sense to answer like "Due to your suggestion, I found some shortcomings in my current work. I will improve the level of scientific research and achieve more results according to your suggestion in the future work."

3) Fig. 4 is meaningless. It can be either removed or changed to supp.

4) Figs 2&3 can be combined. The results are still not such interesting though.